# Salicylic Acid Modulates the Osmotic System and Photosynthesis Rate to Enhance the Drought Tolerance of *Toona ciliata*

**DOI:** 10.3390/plants12244187

**Published:** 2023-12-18

**Authors:** Qi Gao, Yamin Liu, Yumin Liu, Chongwen Dai, Yulin Zhang, Fanbo Zhou, Yating Zhu

**Affiliations:** 1College of Resources and Environment, Southwest University, Chongqing 400715, China; 18883779701@163.com (Q.G.); yaminliu0511@163.com (Y.L.); dchongw@163.com (C.D.); 13364027640@163.com (Y.Z.); scott_chou@163.com (F.Z.); 15187476117@163.com (Y.Z.); 2Key Laboratory of Ecological Environment in Three Gorges Reservoir Area, Ministry of Education, Southwest University, Chongqing 400715, China

**Keywords:** drought tolerance, osmotic system, photosynthesis rate, salicylic acid, soluble proteins

## Abstract

*Toona ciliata* M. Roem. is a valuable and fast-growing timber species which is found in subtropical regions; however, drought severely affects its growth and physiology. Although the exogenous application of salicylic acid (SA) has been proven to enhance plant drought tolerance by regulating the osmotic system and photosynthesis rate, the physiological processes involved in the regulation of drought tolerance by SA in various plants differ. Therefore, drought mitigation techniques tailored for *T. ciliata* should be explored or developed for the sustainable development of the timber industry. We selected 2-year-old *T. ciliata* seedlings for a potting experiment, set the soil moisture at 45%, and subjected some of the *T. ciliata* seedlings to a moderate drought (MD) treatment; to others, 0.5 mmol/L exogenous SA (MD + SA) was applied as a mitigation test, and we also conducted a control using a normal water supply at 70% soil moisture (CK). Our aim was to investigate the mitigating effects of exogenous SA on the growth condition, osmotic system, and photosynthesis rate of *T. ciliata* under drought stress conditions. OPLS–VIP was used to analyze the main physiological factors that enable exogenous SA to alleviate drought-induced injury in *T. ciliata*. The results indicated that exogenous SA application increased the growth of the ground diameter, plant height, and leaf blades and enhanced the drought tolerance of the *T. ciliata* seedlings by maintaining the balance of their osmotic systems, improving their gas exchange parameters, and restoring the activity of their PSII reaction centers. The seven major physiological factors that enabled exogenous SA to mitigate drought-induced injury in the *T. ciliata* seedlings were the soluble proteins (Sp), net photosynthetic rate (*P*n), transpiration rate (*T*r), stomatal conductance (*G*s), stomatal opening window (*S*ow), activity of the photosystem II reaction center (ΦPSII), and electron transfer rate (ETR). Of these, Sp was the most dominant factor. There was a synergistic effect between the osmotic system and the photosynthetic regulation of drought injury in the *T. ciliata* seedlings. Overall, our study confirms that exogenous SA enhances the drought tolerance of *T. ciliata* by modulating the osmotic system and photosynthesis rate.

## 1. Introduction

Abiotic stress negatively affects plant growth and development, resulting in significant losses to the agroforestry sector worldwide [1]. Drought stress is the most critical abiotic factor that limits plant growth, distribution, and survival, thereby significantly hampering forest productivity [2,3]. Drought limits plant growth and development through different mechanisms, such as the inhibition of photosynthesis and the disruption of the osmoregulatory systems [4,5,6,7]. As global warming intensifies, it is projected that more than 50% of the earth will experience water scarcity by 2050 [8]; thus, plants are subjected to an increasingly arid and hot environment. Therefore, methods for mitigating drought damage in plants should be explored or developed.

Plants produce antioxidants, induce primary or secondary metabolites, and undergo physiological and metabolic responses to stressors as their mechanisms to tolerate drought stress [9,10,11,12]. Muhammad showed that plants can cope with drought-induced damage by maintaining their water balance through stomatal transpiration [13]. Stomata are channels through which plants exchange gases with the environment; their opening and closing directly determine the photosynthetic rate of plants [14]. However, these natural drought defense mechanisms have limited effectiveness. Chemical modulation techniques, including the use of salicylic acid (SA)—a phenolic compound that modulates various physiological and biochemical processes to enable plants to cope with environmental stresses [15,16]—have been proven to mitigate drought damage in plants [17]. Specifically, SA modulates the osmotic system and the photosynthetic capacity of plants during drought stress [18,19]. For instance, exogenous SA application successfully mitigated the negative effects of drought on kale by increasing its proline and antioxidant enzyme content and decreasing its membrane lipid peroxidation [20]. It also alleviated drought damage in sweet potatoes by increasing their rate of photosynthesis [21]. However, the physiological mechanisms behind the modulatory effects of SA on drought tolerance differ among plant species, and the synergistic effects of different physiological processes are unclear. Furthermore, research on the dominant SA-induced factors that enhance drought tolerance in plants is scarce. These shortcomings affect our overall understanding of the regulation of drought damage by SA in plants and hamper the development of strategies that employ SA to modulate drought damage in plants.

*Toona ciliata* M. Roem., a tall tree species in the *Meliaceae* Juss. family, is classified as ‘Vulnerable’ in *The IUCN Red List of Threatened Species*. It is mainly distributed in Southeast Asia and is highly prized as a fast-growing and valuable timber tree species in subtropical China [22]. The heartwood of *T. ciliata* is dark reddish brown with straight grain, a detailed structure, tough texture, and beautiful pattern, and it is known as the “Chinese mahogany” [23]. It is a great material for construction, furniture, and interior decoration. *T. ciliata* is highly sensitive to water availability, and an adequate water supply is the basis for the maintenance of its normal growth. Drought negatively affects its physiological and biochemical processes, resulting in retarded growth and the lowering of its yield and quality [24]. As global warming intensifies, variable precipitation and seasonal droughts pose a serious threat to the growth and survival of water-demand-sensitive trees in subtropical China [25], such as *T. ciliata*. Therefore, methods for mitigating drought damage in *T. ciliata* should be explored for the development of management strategies aimed at maintaining the long-term productivity of *T. ciliata* in arid environments.

The objectives of this study were to (1) analyze how exogenous SA enhances drought resistance in *T. ciliata* by modulating its osmotic system and photosynthesis rate; (2) determine the physiological processes and factors that enable exogenous SA to alleviate drought damage in *T. ciliata*; and (3) determine whether these factors elicit synergistic regulatory effects. To attain these objectives, we subjected 2-year-old *T. ciliata* seedlings to drought stress in a pot experiment and sprayed them with 0.5 mmol/L SA. The findings of this study could be useful in developing management strategies for the long-term productivity maintenance of *T. ciliata* in arid environments.

## 2. Results

### 2.1. Growth and Morphology

The growth of the *T. ciliata* seedlings was inhibited by drought, as evidenced by the significantly reduced ground diameter, plant height, and leaf length and width values (Table 1, *p* < 0.05). Following SA application, the ground diameter, plant height, and leaf length and width of the *T. ciliata* seedlins significantly improved (*p* < 0.05).

Drought stress significantly reduced the total chlorophyll (Chlt) content of the leaves (*p* < 0.001), as evidenced by the light yellow leaf color, the presence of white spots, the reddening of the leaf veins, and the wilting and shedding of the leaf blades of the *T. ciliata* seedlings. After they were sprayed with SA, the Chlt content of the leaves of the MD + SA group increased compared with those in the MD group (*p* < 0.05); moreover, leaf color was restored and leaf wilting was reduced (Figure 1, Table 2).

### 2.2. Osmotic Substances

Apart from a non-significant alteration (*p* > 0.05) in the soluble sugar (Ss) content of the MD + SA group (Figure 2A), the exogenous application of SA had a significant regulatory effect on the osmotic substances in the *T. ciliata* seedlings. Under drought stress, the proline (Pro) content of the *T. ciliata* seedlings significantly increased (Figure 2B), suggesting inherent drought tolerance; it increased further after the SA was sprayed (*p* < 0.001). Under drought stress, the starch (St), soluble protein (Sp), and leaf water content (LWC) of the *T. ciliata* seedlings significantly decreased (Figure 2C–E), and the mesophyll cells lost water and plasmolyzed (Figure 2F). After the exogenous application of SA, the St, Sp, and LWC of the *T. ciliata* seedlings significantly increased (*p* < 0.001), the intracellular environment was improved, and the osmotic system attained homeostasis. Interestingly, exogenous SA partially restored the morphology of the leaf mesophyll cells under drought conditions. However, a significant discrepancy remained when compared to the control group. This could be attributed to the activation of the plant’s defense system in response to drought, leading to an accumulation of reactive oxygen species (ROS). The inability of SA to quickly scavenge these ROS might have contributed to the limited restoration of the mesophyll cells within a short timeframe [26].

### 2.3. Photosynthetic Index

#### 2.3.1. Gas Exchange Parameters

Under drought stress, the gas exchange parameters of the *T. ciliata* seedlings were inhibited (Figure 3): the photosynthetic rate (*P*n), transpiration rate (*T*r), intercellular CO_2_ concentration (*C*i), stomatal conductance (*G*s), and stomatal opening width (Sow) values significantly decreased, while the stomatal limiting (*L*s) and water use efficiency (WUE) significantly increased (Figure 3A–G). Moreover, the stomata decreased in size (Figure 3H). After the SA was sprayed, the *P*n, *T*r, *G*s, and *S*ow significantly increased (*p* < 0.001), the *L*s decreased significantly (*p* < 0.001), the stomata returned to normal (Figure 3H), and the Ci and WUE slightly increased (*p* > 0.05) compared with the MD group. Overall, exogenous SA application had a significant regulatory effect on the gas exchange parameters of the *T*. *ciliata* seedlings under drought stress.

#### 2.3.2. Chlorophyll Fluorescence Parameters

The chlorophyll fluorescence parameters reflect the activity of the PSII reaction center in *T. ciliata*. Under drought stress, the initial fluorescence (*F*o) and non-photochemical quenching (NPQ) of the *T. ciliata* seedlings significantly increased (*p* < 0.001); the maximal photochemical efficiency of the PSII (*F*v/*F*m), the actual photochemical efficiency (ΦPSII), the photochemical quenching (*q*P), and the electron transfer rate (ETR) significantly decreased (*p* < 0.001); and the PSII reaction center was inactivated (Figure 4). After the SA was sprayed, all of the chlorophyll fluorescence parameters of the *T. ciliata* seedlings were restored to varying degrees: for instance, the ΦPSII and ETR were significantly restored (*p* < 0.001), and the PSII reaction center was reactivated.

### 2.4. Major Regulatory Mechanisms of Exogenous SA Affecting Drought Injury in T. ciliata

Nineteen physiological indicators were evaluated using PCA (Figure 5), which explained 95.48% of the variations in the data. Specifically, principal component PC1 was responsible for the majority of the photosynthetic indicators and the Sp content, accounting for 63.97% of the variation, whereas PC2 predominantly explained the osmotic substances, excluding Sp, accounting for 31.51% of the variation. MD and MD + SA were clearly separated by PC2, and MD + SA was positively correlated with osmotic substances and most photosynthetic indicators except Ss, and negatively correlated with Ls and Fo.

To further identify the primary physiological factors that enable SA to alleviate drought stress in *T. ciliata* and reveal the regulatory mechanisms of SA, we conducted a VIP analysis using the OPLS model (Figure 6) to calculate the relationship between the VIP values of 19 physicochemical indices and the ground diameter, plant height, and leaf length and width of the *T. ciliata* seedlings. Among these indices, Sp, *P*n, *T*r, *G*s, *S*ow, ΦPSII, and ETR had VIP values greater than 1.0; Sp had the largest VIP value and the most sensitive response to drought stress. These seven indicators significantly influenced the *T. ciliata* growth and were the main physiological factors that alleviated drought stress in the *T. ciliata* seedlings through the action of SA. Among these indicators, Sp had the most dominant influence.

### 2.5. Synergistic Regulation among the Major Factors

Of the seven main physiological factors (VIP > 1.0) that alleviated drought damage in the *T. ciliata* seedlings through the action of SA, only one was an osmotic regulator (Sp), and the remaining six were photosynthetic regulators. A highly significant positive correlation was observed between Sp content and the other six photosynthetic parameters (Figure 7), suggesting a synergistic relationship between osmotic and photosynthetic regulation that enabled the SA to mitigate drought-induced injury in the *T. ciliata* seedlings.

## 3. Discussion

### 3.1. Exogenous SA Regulates the Apparent Morphology of T. ciliata under Drought Stress

Under drought stress, plant cell division was inhibited, and biomass allocation tended to favor the underground parts, resulting in short and slow growth of the aboveground parts of the plant and the retarded growth and development of various organs [27]. In this study, the growth rate of all parts of the *T. ciliata* seedlings decreased under drought stress, and the leaf color became yellow and light, indicating that drought conditions limit the growth of *T. ciliata*. After SA application, the apparent morphology and chlorophyll content of the *T. ciliata* seedlings were restored to varying degrees. This is because SA is a phenolic compound, and phenolics can act both as regulators of stomatal movement and with growth regulators [28], and the exogenous spraying of SA can stimulate the metabolism of growth regulators, such as indole acetic acid (IAA) and ethylene (CH_2_) [29], thereby alleviating drought-induced inhibitions on the growth and development of *T. ciliata*. In addition, the application of SA can increase both the content of photosynthetic pigments in plants and the accumulation of α-aminolevulinic acid (α-ALA), which is an intermediate in chlorophyll synthesis [30] that promotes the synthesis of chlorophyll in the leaves of *T. ciliata*. This stimulated the drought-inhibited photosynthetic processes in the *T. ciliata* seedlings and restored the color of the leaf blades to their normal shade.

### 3.2. Exogenous SA Modulates the Osmotic System to Enhance the Drought Tolerance of T. ciliata

The osmotic system is the primary defense mechanism of plants under drought stress; it plays a vital role in maintaining cellular water and expansion pressure [31]. The uptake of mineral elements is reduced under drought stress, and osmoregulation mainly occurs through the action of small organic molecules [32]. In this study, drought conditions disrupted the osmotic system of the *T. ciliata* seedlings, resulting in water loss from mesophyll cells, a significant decrease in St and Sp levels, and a significant increase in Pro levels (Figure 2). An increased Pro level is a defense mechanism that *T. ciliata* employs to cope with drought stress. The St and Sp content of the *T. ciliata* seedlings significantly increased after they was sprayed with SA, whereas their Pro content increased further compared with that of the seedlings in the MD group. This may be due to the induction of proline synthase gene expression by the SA [33]. Elevated Pro content increases the levels of intracellular osmotic substances and ensures cellular water balance, ultimately alleviating drought-induced damage in *T. ciliata*. Interestingly, the exogenous SA partially restored the morphology of the leaf mesophyll cells under drought conditions. However, a significant discrepancy remained when these seedlings were compared to those in the control group (Figure 2F). This could have been a result of the activation of the plants’ defense system in response to drought, which may have led to the accumulation of reactive oxygen species (ROS). The inability of SA to quickly scavenge these ROS might have contributed to the limited restoration of the mesophyll cells within a short timeframe [26]. Therefore, exogenous SA can improve the intracellular environment and maintain the balance of the osmotic system to a certain extent, and the OPLS–VIP results indicated that alterations in the Sp content had a significant impact on the ground diameter, plant height, and leaves (Figure 7). Sp stores and releases nutrients, such as nitrogen and phosphorus, to maintain normal plant growth and development [34], and it plays an important role in plant responses to environmental stress. It was found in this study that drought stress caused insufficient water to reach the root system of the *T. ciliata* seedlings and decreased their nitrogen uptake capacity. Nitrogen is a fundamental component of proteins, and nitrogen content affects Sp synthesis; hence, the Sp content in the *T. ciliata* leaves significantly decreased under drought stress. Exogenous SA application promotes the expression of Sp genes, particularly the heat shock protein family [35], which rapidly aggregate in the presence of SA and elicit protective effects [36]. In addition, exogenous SA application can increase the synthesis of various nonspecific proteins, including Pro [37], and enhance Sp stability, thereby maintaining the intracellular environment in response to drought stress. Therefore, exogenous SA application can alleviate drought-induced damage in *T. ciliata* by maintaining the balance of the osmotic system, with Sp being the dominant factor.

### 3.3. Exogenous SA Modulates Photosynthesis to Enhance Drought Tolerance in T. ciliata

In this study, the *T. ciliata* seedlings, like most leafy plants [38,39], exhibited a certain degree of drought tolerance and coped with the drought stress by reducing their *G*s and increasing their WUE. However, this self-defense mechanism is insufficient for coping with long-term drought damage. SA spraying significantly increased the *G*s and *S*ow values of the *T. ciliata* seedlings. As a signaling molecule [40], SA can improve the drought resistance of *T. ciliata* by regulating the K^+^ channels to open the stomata [41], facilitating the ion exchange process and increasing the cellular osmotic concentration [42]. Exogenous SA application increased the *T*r of the *T. ciliata* seedlings; consequently, the transpiration pull of the *T. ciliata* seedlings increased, enabling them to absorb more nutrients from the soil [43] and promoting their growth and development. In addition, elevated *T*r can regulate the surface temperature of *T. ciliata* leaves [44], enabling them to avoid overheating and leading to improved drought tolerance in *T. ciliata*.

Chlorophyll fluorescence is an important photosynthetic parameter for light energy absorption and utilization [45]. In this study, SA spraying significantly increased the ΦPSII and ETR of the *T. ciliata* seedlings, resulting in an increased electron transfer rate and restored activity in the PSII reaction center, which ultimately led to increased photosynthesis [46]. Therefore, we suggest that exogenous SA improves the drought tolerance of *T. ciliata* by promoting photosynthesis via two pathways: the regulation of photosynthetic parameters and the restoration of the activity of the PSII reaction center [47].

### 3.4. Synergistic Physiological Processes of Exogenous SA which Enhance Drought Tolerance in T. ciliata

Under drought stress, exogenous SA application enhanced the ground diameter, plant height, and leaf growth of the *T. ciliata* seedlings, diminished leaf wilting, restored leaf color, and mitigated the negative impacts of drought on the *T. ciliata* seedlings by regulating their osmotic systema and photosynthesis rates. Based on the OPLS–VIP results (Figure 6), we identified seven major synergistic factors—with Sp being the dominant factor—that enabled exogenous SA application to alleviate drought injury in the *T. ciliata* seedlings. The results of a linear correlation analysis (Figure 7) revealed that Sp was significantly correlated with both gas exchange parameters and activity in the PSII reaction center. We attribute the increase in the Sp content, *G*s, and *S*ow of the *T. ciliata* seedlings after exogenous SA application to the fact that soluble proteins increase the intracellular osmotic substance content, restores normal osmotic pressure, and affects the degree to which the stomata open and close [48]. Soluble proteins contain elements such as nitrogen and sulfur, which provide plants with essential nutrients and metabolites and thus maintain the integrity of the stomatal structure and function and restore the original permeability [49]. Soluble proteins can also regulate the cell wall composition and the activity of related enzymes, which promote the expansion and growth of the stomatal cell wall and maintain the normal function of the stomatal structure [50,51]. Moreover, Sp content was significantly correlated with the activity of the PSII reaction center of the *T. ciliata* seedlings (Figure 7E,F). Studies have also demonstrated that soluble proteins can bind to both electron donors and acceptors [52] while performing functions such as protecting the light response complex and maintaining biofilm stability [53]. For example, some soluble proteins bind to chlorophyll proteins, which regulate the rate of electron uptake and release from the light reaction centers [54]. In summary, exogenous SA application improved the drought tolerance of the *T. ciliata* seedlings by regulating their osmotic systems and photosynthesis rates; Sp content was the factor that had the most important effect on the stomata and PSII reaction centers of the *T. ciliata* seedlings, resulting in increased photosynthesis rates and the alleviation of drought-induced damage (Figure 8).

## 4. Materials and Methods

### 4.1. Source of Plants

At the beginning of July 2019, healthy 1-year-old *T. ciliata* seedlings with ground diameters of 0.6 cm and heights of approximately 1 m were transplanted to the test base for potting trials. A nutritional bowl with a 25 cm caliber and a 20 cm depth was selected, and each bowl was filled with 5 kg of culture medium—a mixture of soil/peat soil (2:1 by volume). The bowls were uniformly fitted with potting mats and watered normally. In July 2020, a year after the seedlings had rooted in the pots, healthy and uniform *T. ciliata* seedlings were thoroughly watered and placed in a rain-proof shelter for testing.

### 4.2. Treatment and Training Conditions of Test Materials

The Chinese National Standard of Meteorological Drought Grade (GB/T 20481-2017) [55] was followed in this study, and the median value of the water control interval for moderate drought (40% < soil relative humidity ≤ 50%) was 45%. Based on the pre-test results obtained using a single-factor concentration gradient, we determined that 0.5 mmol/L SA was the most appropriate concentration for the subsequent drought stress mitigation tests. The experiment involved three groups: a normal control group that received a normal amount of water (CK), a drought control group (MD), and an SA-conditioned group (MD + SA). Each group consisted of three plants and was replicated five times. Before the onset of drought stress, the adaxial and abaxial sides of the leaves of the seedlings in the MD + SA group were sprayed with 0.5 mmol/L SA solution daily at 6 pm; the leaves were thoroughly moistened without allowing any water droplets to drip. The MD and CK groups were treated with pure water. After three consecutive days of spraying, the groups were watered abundantly, and natural drought was initiated for all except the CK group. The soil relative humidity was monitored using a Delta-T wet sensor (Beijing Bolun Jingwei Technology Development, Beijing, China). Water control was initiated when the soil relative humidity reached 45 ± 3% in the MD and MD + SA groups, and the MD + SA group was supplemented with one spray of SA solution. Subsequently, the weighing method was employed to maintain the soil relative humidity at 45 ± 3% in the MD and MD + SA groups and at 70 ± 3% in the CK group. The indices were measured after 9 days (Figure 9).

### 4.3. Growth and Morphology

Two index measurements were performed (with a 27-day interval), the first before the 1st application of the SA solution, and the second 9 days after the induction of drought stress (Figure 9). The ground diameters of the *T. ciliata* seedlings were measured using a Vernier caliper. The selected measurement site was proximal to the base of the soil surface and was marked with a marker pen to ensure that the same seedling position was used when the two sets of measurements were taken. A soft ruler was used to measure the height of each plant from the soil surface to the shoot apex. Functional leaves from the 3rd to the 5th leaf sequence from the new shoots were selected, and the leaf lengths and widths were measured using a Vernier caliper. The difference between the two sets of measurements was calculated as the increase in each index.

After 9 days of drought stress, we observed the morphology of the plants, including the number of leaves, degree of wilting, leaf color, and foliar features (Figure 9). We used a Nikon somatic microscope (C-LEDS; Nikon, Tokyo, Japan) to observe and take photographs of the leaf colors. We used a Leica DM1000LED biomicroscope (Leica Camera, Wetzlar, Germany) to observe and take photographs of the stomatal morphology and chloroplasts. Sow was measured using “MinFeret” in Image J2 software.

### 4.4. Osmotic Substances

After 10 days of drought stress, intact functional leaves from the 3rd to the 5th leaf sequence of the new shoots in each group of plants were collected for osmotic substance determination (Figure 9), and some were stored in liquid nitrogen for backup.

The LWC values of the leaves was measured using the drying and weighing method [56]. First, the initial mass *m* was determined; the leaves were then placed in a paper bag and dried until a constant weight was achieved, and the dried mass *m′* was recorded. The LWC values were calculated using the formula below.
(1)$$LWC=\frac{m-m’}{m\;}\times 100\%$$

The Ss and St content of the leaves was measured using the anthrone colorimetric method [57].

The Sp content of the leaves was measured by employing the Coomassie brilliant blue G-250 method [58], and the Pro content of the leaves was measured using the method described by Bates et al. [59].

### 4.5. Photosynthetic Index

After 9 days of drought stress, the functional leaves from the 3rd to the 5th leaf sequence of the new shoots were selected, and the gas exchange parameters were determined using a Portable Photosynthesis System (LI-6800, Li-Cor, Nebraska, USA, manufactured by LigaoTai Technology in Beijing, China) (Figure 9). The CO_2_ concentration was set at 380 μmol/mol, and the temperature was set at 35 °C. The leaf temperature was set at 25 ± 1 °C, the light intensity at 1,800 μmol/m^2^·s, the gas flow rate at 500 μmol/s, and the relative humidity at approximately 60–70%. The measured indices were the *P*n, the *T*r, the atmospheric CO_2_ concentration (*C*a), the *C*i, and the *G*s. These indices were used to calculate the WUE and *L*s as shown in the equations below.
(2)WUE=Pn÷Tr
(3)$$Ls=\frac{Ca-Ci}{Ca\;}\times 100\%$$

After 10 days of drought stress, the chlorophyll fluorescence parameters were measured using a Portable Photosynthesis System (LI-6800, Li-Cor, Nebraska, USA, manufactured by LigaoTai Technology in Beijing, China) (Figure 9). The functional leaves from the 3rd to the 5th leaf sequence of the new shoots were wrapped in tin foil overnight to determine the *F*o, and the maximal fluorescence (Fm) was measured at an exposure time of 0.8 s under saturated pulsed light at 5000 μmol/m^2^·s. Afterwards, the leaves were acclimatized at a light intensity of 800 μmol/m^2^·s for 30 min to determine the maximal fluorescence (*F*m′). The minimum fluorescence (*F*o′) was determined using the ‘Dark Pulse’ program of the Li-6800 photosynthesizer. The remaining parameters, including the variable fluorescence (*F*v), *F*v/*F*m, ΦPSII, NPQ, *q*P, and ETR, were calculated using the photosynthesizer.

After 10 days of drought stress, the leaves were collected to measure their chlorophyll content. A 6 mm diameter disc was punched out of the leaves and immersed in a mixture of 95% alcohol and acetone (1:1). Chlorophyll was extracted in the dark for 24 h until the leaves turned white. The chlorophyll a (Chla), chlorophyll b (Chlb), and Chlt content per unit leaf area values were calculated at 645, 652, and 663 nm, respectively; 80% acetone was used as the control.

### 4.6. Statistical Analysis

The means and standard deviations for each treatment are presented. To compare the data for each treatment, a one-way analysis of variance was performed using SPSSS version 19.0. A principal component analysis (PCA) was performed on 19 physiological indicators using CANOCO 5 to preliminarily determine the main physiological indicators affecting the growth of the *T. ciliata* seedlings. Using SIMCA 14.1, the variable influence on projection (VIP) values of the 19 physicochemical indicators were quantified using a VIP analysis in the OPLS model in relation to the ground diameters, plant heights, and leaf lengths and widths of the *T. ciliata* seedlings. OPLS is a regression modelling method with multiple dependent variables on multiple independent variables. Its most important feature is that it can remove data variance in the independent variable X that is not related to the categorical variable Y; hence, the categorical information is mainly concentrated in one principal component [60]. OPLS can achieve effective separation and construct a categorical prediction model that can be used to identify more sample categories, thereby making the model more explanatory [61]. The VIP values allowed us to identify the dominant physiological indicators that influence the ground diameter, plant height, leaf length, and leaf width of *T. ciliata*. A VIP value greater than 1.0 indicates that the tested indicator has a strong influence on growth. Based on the results of the OPLS–VIP analysis, we selected the physiological indicators by which exogenously applied SA had a strong influence (VIP > 1.0) on the growth of *T. ciliata* and analyzed the correlations between the indicators using linear correlation.

## 5. Conclusions

In conclusion, exogenous SA application can mitigate drought-induced injury in *T. ciliata* by maintaining the balance of its osmotic system, improving its gas exchange parameters, and restoring the activity of its PSII reaction center. Sp, *P*n, *T*r, *G*s, *S*ow, ΦPSII, and ETR were the seven major physiological factors that enabled exogenous SA to alleviate drought-induced injury in the *T. ciliata* seedlings; Sp was the most dominant factor. There is synergy between the osmotic and photosynthetic regulation of drought damage in *T. ciliata*, and Sp had important effects on stomatal opening and the activity of the PSII reaction center.

Our study confirmed that exogenous SA application modulates the osmotic system and photosynthesis rate of *T. ciliata*. Our findings are useful for the development of management strategies for the maintenance of the long-term productivity of *T. ciliata* in arid environments.

## Figures and Tables

**Figure 1 plants-12-04187-f001:**
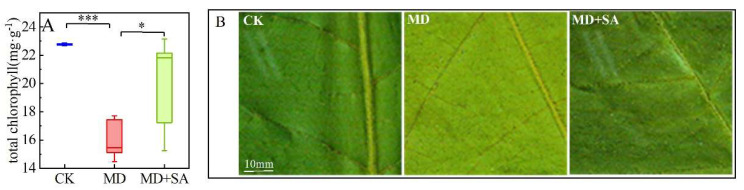
Alleviating effects of exogenous SA on chlorophyll content and leaf color of *T. ciliata* seedlings under drought stress. Chlorophyll content changes (**A**). Leaf color changes (**B**). * indicates *p* < 0.05, *** indicates *p* < 0.001. The same is true in the figures below.

**Figure 2 plants-12-04187-f002:**
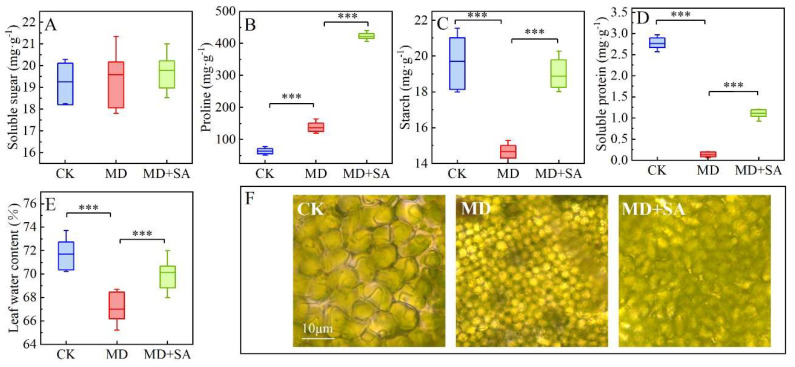
Effects of exogenous SA on osmotic regulator in leaves of *T. ciliata* seedlings under drought stress. Osmotic adjustment substance changes (**A**–**E**). Morphological changes to mesophyll cells (**F**). *** indicates *p* < 0.001.

**Figure 3 plants-12-04187-f003:**
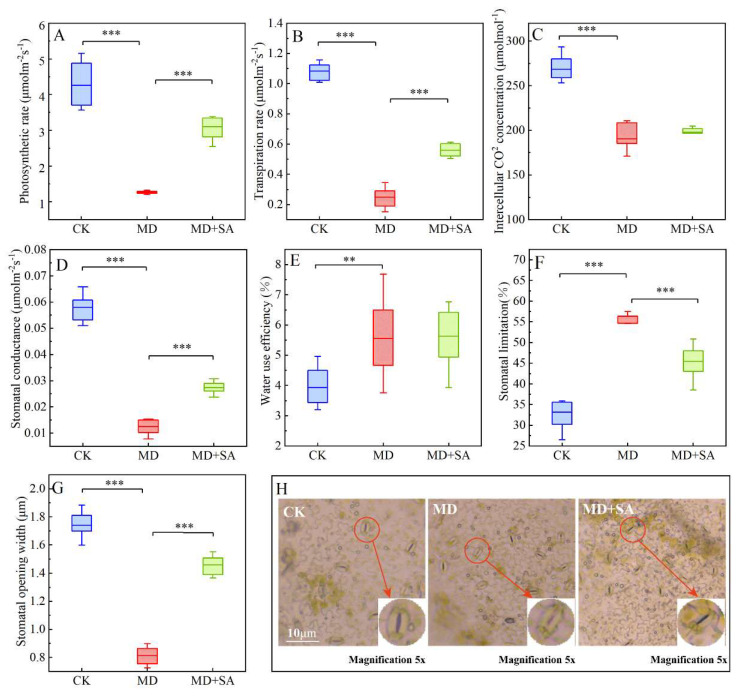
Effects of exogenous SA on gas exchange parameters in leaves of *T. ciliata* seedlings under drought stress. Gas exchange parameter changes (**A**–**G**). Stomatal morphological changes (**H**); ** indicates *p* < 0.01, *** indicates *p* < 0.001.

**Figure 4 plants-12-04187-f004:**
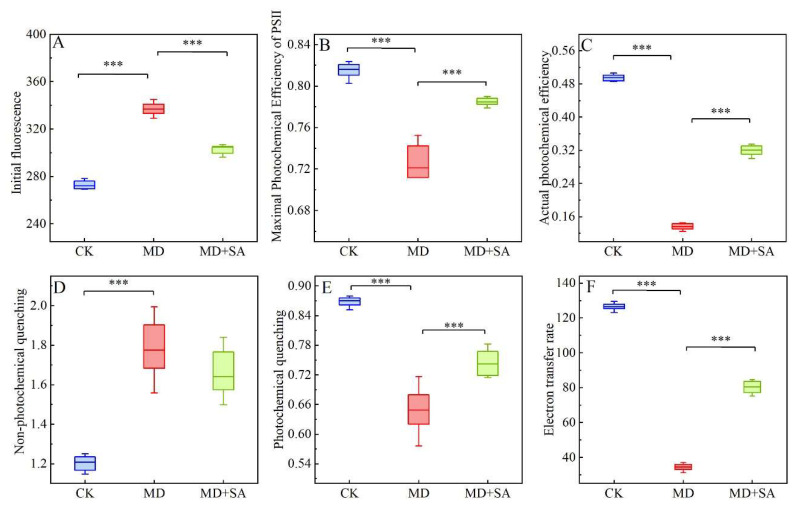
Effects of exogenous SA on chlorophyll fluorescence of *T. ciliata* seedlings under drought stress. Initial fluorescence changes (**A**). Maximal photochemical efficiency of PSII changes (**B**). Actual photochemical efficiency changes (**C**). Non-photochemical quenching changes (**D**). Photochemical quenching changes (**E**). Electron transfer rate changes (**F**). *** indicates *p* < 0.001.

**Figure 5 plants-12-04187-f005:**
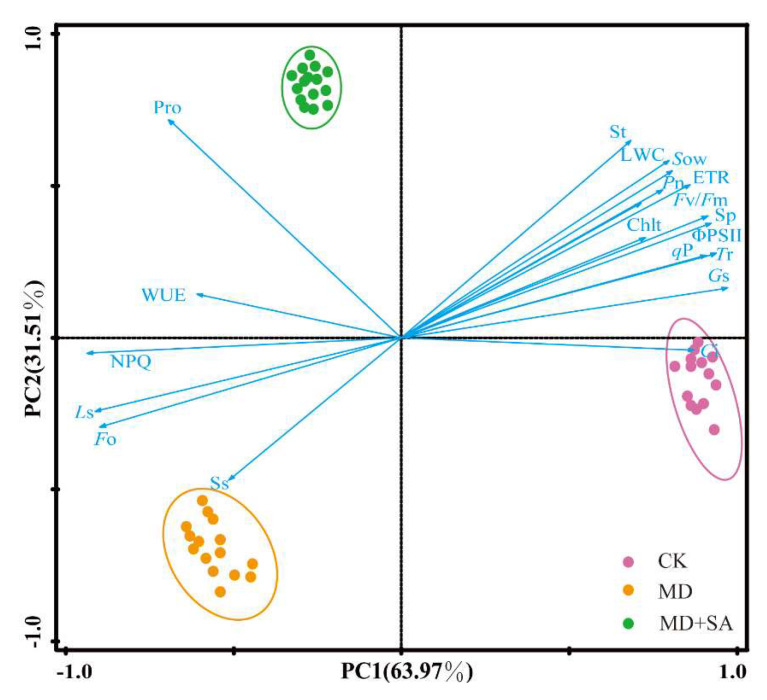
PCA analysis of physiological indices of *T. ciliata* seedlings after exogenous SA application.

**Figure 6 plants-12-04187-f006:**
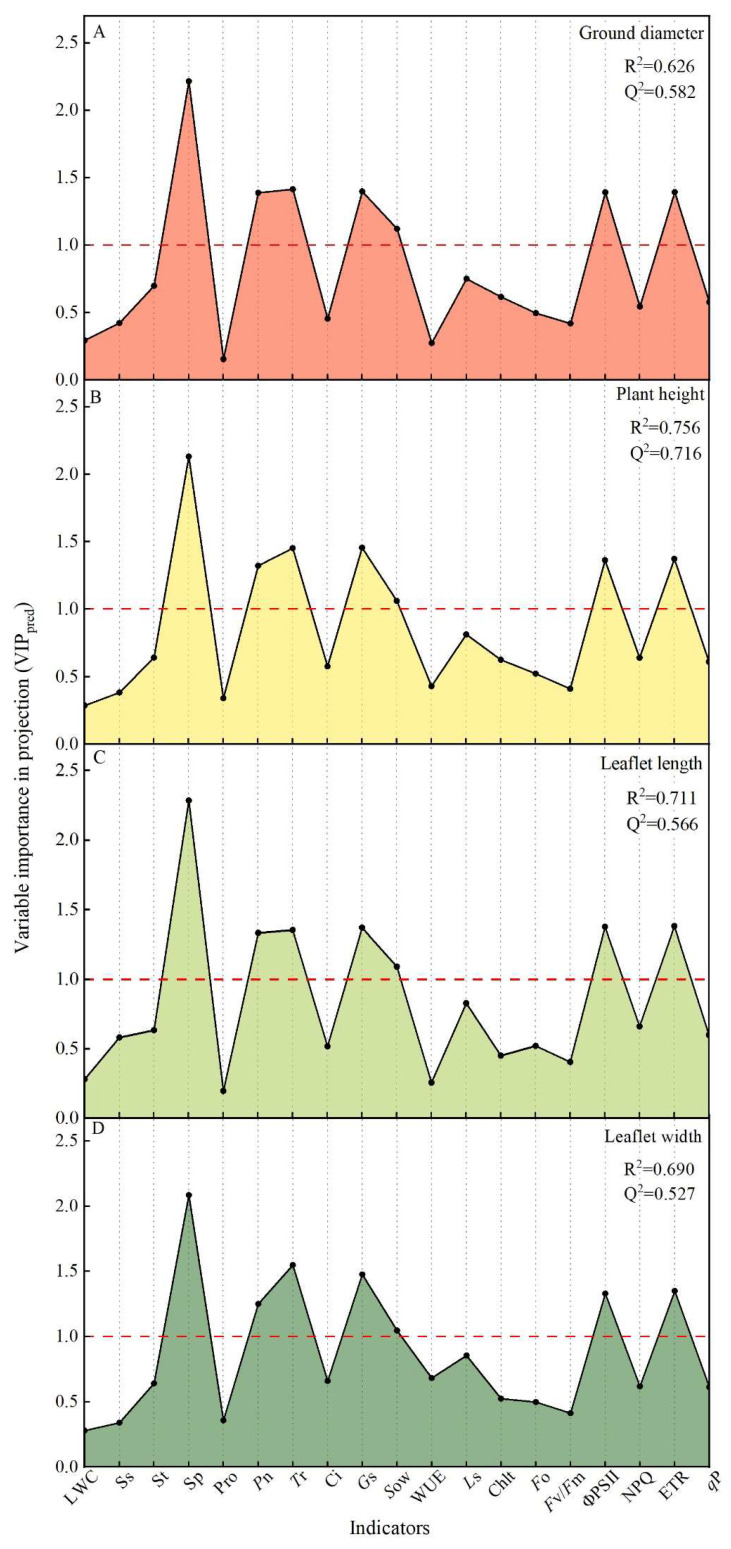
The OPLS model was used to obtain predicted VIP values for each index. The study identified the primary physiological factors that regulate the ground diameter (**A**), plant height (**B**), leaf length (**C**), and leaf width (**D**) in *T. ciliata* seedlings when SA is applied exogenously. R^2^, model goodness of fit; Q^2^, prediction accuracy of the model. R^2^ and Q^2^ values higher than 0.4 are acceptable, and values higher than 0.5 are better.

**Figure 7 plants-12-04187-f007:**
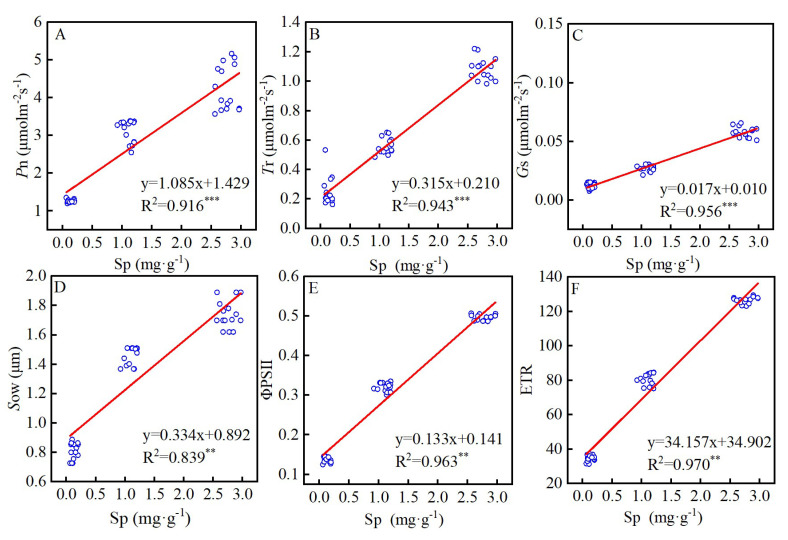
Correlations between Sp and photosynthetic parameters. Photosynthetic rate (**A**). Transpiration rate (**B**). Stomatal conductance (**C**). Stomatal opening width (**D**). Actual photochemical efficiency (**E**). Electron transfer rate (**F**). The red lines represent the trend lines; ** *p* < 0.01; *** *p* < 0.001.

**Figure 8 plants-12-04187-f008:**
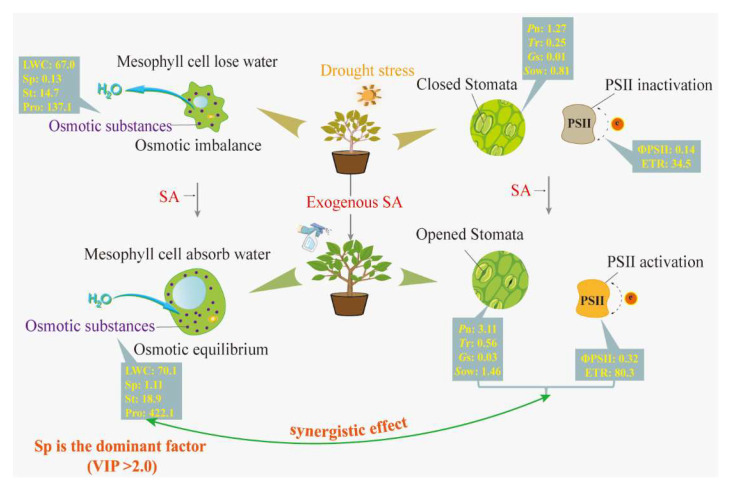
Physiological processes by which exogenous SA mitigates drought injury in *T. ciliata*. Exogenous SA can improve the drought tolerance of *T. ciliata* by regulating the osmotic system and photosynthesis. Sp, *P*n, *T*r, *G*s, *S*ow, ΦPSII, and ETR were the main physiological factors, and there was a synergistic regulation among them. Sp was the dominant factor.

**Figure 9 plants-12-04187-f009:**
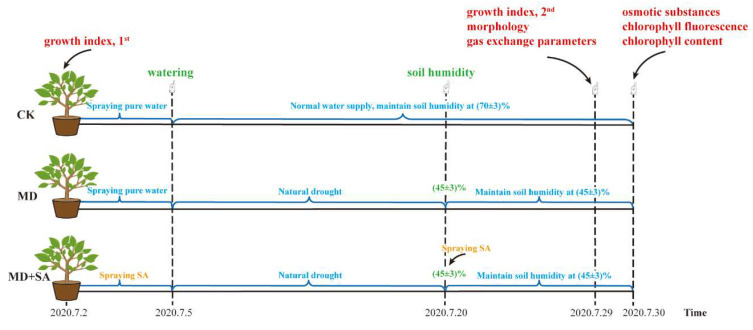
Schematic diagram of the experimental procedure.

**Table 1 plants-12-04187-t001:** Effect of exogenous SA on the growth rate of *T. ciliata*.

Treatment	Ground Diameter Growth Rate (%)	Height Growth Rate (%)	Leaflet Length Growth Rate (%)	Leaflet Width Growth Rate (%)
CK	5.67 ± 0.44 a	4.59 ± 0.51 a	2.70 ± 0.39 a	10.02 ± 2.18 a
MD	4.80 ± 1.50 b	2.78 ± 0.56 b	1.88 ± 0.24 b	6.05 ± 2.21 b
MD + SA	5.64 ± 0.59 a	4.42 ± 0.53 a	2.63 ± 0.47 a	9.32 ± 2.44 a

Values are X ± SD. Different lowercase letters after data in the same column indicates significant differences between different treatments (*p* < 0.05).

**Table 2 plants-12-04187-t002:** Alleviating effects of exogenous SA on the morphology of *T. ciliata* seedlings under drought stress.

Treatment	Leaf Color	Degree of Leaf Wilting	Morphological Description
CK	A	C	The trees are straight and dark green, spotless and wilting. The tree is straight; its leaves are the lightest and they have white spots.
MD	C	A	The tips of the leaflets are dry, the veins are red, and the petioles turn yellow and fall off from below, leaving only four rounds of compound leaves at the top.
MD + SA	B	B	The tree body is straight, the leaves are thicker and larger, without spots, the degree of wilting is lighter, and the bottom leaves become yellow and fall off.

Leaf color and wilting degree are represented by the letters A–C; A was the highest degree, and it decreased successively from A to C.

## Data Availability

The data used to support the findings of this study are available from the corresponding author upon request.

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
