# Peer review of "Salicylic Acid Modulates the Osmotic System and Photosynthesis Rate to Enhance the Drought Tolerance of *Toona ciliata"

_plants, 2023, doi:10.3390/plants12244187_

Round 1
Reviewer 1 Report
Comments and Suggestions for Authors
Report about:
Salicylic Acid Modulates the Osmotic System and Photosynthe-sis to Enhance the Drought Tolerance of Toona ciliata
2750299
The aim of the paper were to (1) analyze how exogenous SA enhance drought resistance in T. ciliata by modulating its osmotic system and photosynthesis; (2) determine the physiological processes and factors that enable exogenous SA to alleviate drought damage in T. ciliata; and (3) determine whether these factors elicit synergistic regulatory effects. To attain these objectives, the authors subjected 2-year-old T. ciliata seedlings to drought stress in a pot experiment and sprayed them with 0.5 mmol/L SA. The findings of this study would be useful in developing management strategies for the long-term productivity maintenance of T. ciliata in arid environments. The study confirmed that exogenous SA enhances the drought tolerance of T. ciliata by modulating the osmotic system and photo-synthesis.
This is a very well written review paper, and I enjoyed reading it. This is a very interesting article touching a very hot and applied point. It was a pleasure to read. The figures are excellent, and the tables are also very good but need to be improved as below. This paper contains nicely designed tables and figures. I consider the topic of this paper original and relevant to the field and suitable for the journal. The paper addressed a specific gap in the field. The conclusions are consistent with the evidence and arguments presented and the authors addressed all the main questions posed.
My comments to improve the review:
Please make all tables self-explanatory and do not use any abbreviation in the table footnote or in the table legend. Please make all figures self-explanatory and do not use any abbreviation in the figure footnote or in the figure legend.
All references are appropriate, but I suggest adding more 2022 and 2023 references.
Please change the title. The title is not correct and not easy to follow.
Some other comments.
Keywords
Please arrange all key words in alphabetical order.
Please add new references 2023 please.
Please make all tables self-explanatory.
Please make all tables self-explanatory.
In addition, many scientific names in the references must be written in italic format
In order to improve the quality of the paper update the reference list by adding 2022 and 2023 references.
Please follow the instructions to authors on how they write the reference in the list. For references about textbooks, please add the page numbers of the textbook. Also please add the city of the publisher.
Please add photos for the plants with the different treatments. More section in the introduction is needed about Toona ciliata.
This paper can be accepted with minor revision, but I must revise the paper once more to ensure that all my comments were incorporated.

It is OK. Only minor changes is needed.
Author Response
Dear Reviewer:
Thank you for your letter and for the reviewers’ comments concerning our manuscript entitled “Salicylic Acid Modulates the Osmotic System and Photosynthesis to Enhance the Drought Tolerance of Toona ciliata" (ID: plants-2750299). Those comments are all valuable and very helpful for revising and improving our paper, as well as the important guiding significance to our researches. We have studied comments carefully and have made correction (highlighted) which we hope meet with approval.
Responds to the reviewer’s comments:
Please make all tables self-explanatory and do not use any abbreviation in the table footnote or in the table legend. Please make all figures self-explanatory and do not use any abbreviation in the figure footnote or in the figure legend.
Reply: Thank you for your comment, we have revised all the figures and tables in their entirety.
All references are appropriate, but I suggest adding more 2022 and 2023 references.
Reply: Thank you for your comment, we have added new references.
Please change the title. The title is not correct and not easy to follow.
Reply: Thank you for your comment. Our paper focused on the effects of exogenous SA on the osmotic system and photosynthesis of Toona ciliata under drought stress. After much deliberation, we did not find a more suitable title. Thank you again for your comments.
Some other comments.
Keywords
Please arrange all key words in alphabetical order.
Reply: Thank you for your comment, we have rearranged the keywords in alphabetical order.
Please add new references 2023 please.
Reply: Thank you for your comment, we have added new references as requested.
Please make all tables self-explanatory.
Reply: Thank you for your comment, we have changed all the tables.
Please make all tables self-explanatory.
Reply: Thank you for your comment, we have changed all the tables.
In addition, many scientific names in the references must be written in italic format
Reply: Thank you for your comment. We have changed the scientific names in the references.
In order to improve the quality of the paper update the reference list by adding 2022 and 2023 references.
Reply: Thank you for your comment, we have added new references as requested.
Please follow the instructions to authors on how they write the reference in the list. For references about textbooks, please add the page numbers of the textbook. Also please add the city of the publisher.
Reply: Thank you for your comment, we have added new references as requested.
Please add photos for the plants with the different treatments. More section in the introduction is needed about Toona ciliata.
Reply: Thank you for your comment. We have added the introduction as you requested. We strongly agree that you would like to see photos of the plants from each treatment in the main article. However, we have only photographed the whole, not the individual different treatments, so we can only upload photos of the whole to the supplementary file. We are deeply sorry for our negligence and we will pay attention to this issue in our future research, thank you again for your understanding and valuable comments.

Reviewer 2 Report
Comments and Suggestions for Authors
The objectives of this study were to (1) analyze how exogenous salicylic acid (SA) enhance drought resistance in T. ciliata by modulating its osmotic system and photosynthesis; (2) determine the physiological processes and factors that enable exogenous SA to alleviate drought damage in T. ciliata; and (3) determine whether these factors elicit synergistic regulatory effects. There was conducted a pot experiment using 2-year-old T. ciliata seedlings to determine the mitigating effects of exogenous SA on the growth conditions, osmotic system, and photosynthesis of T. ciliata under drought stress. It was found that exogenous SA application increased the growth of ground diameter, plant height, and leaf blades and enhanced the drought tolerance of T. ciliata by maintaining the balance of the osmotic system, improving the gas exchange parameters, and restor-ing the activity of the PSII reaction centre. The major physiological factors that enabled exogenous SA to mitigate drought-induced injury in T. ciliata were the content of soluble proteins, net photosynthetic rate, transpiration rate , stomatal conductance, activity of the photosystem II, and electron transfer rate. There was a synergistic effect between the osmotic system and photosynthetic regulation of drought injury in T. ciliata.
General comments: Much research has been devoted to assess the effect of salicylic acid on plant response to drought stress. Thus, from this perspective there is no much novelty in this manuscript. Notwithstanding, plants from different species may react differently to effect of salicylic acid on plant response to water deficit, thus, research is still needed on this subject. The manuscript deals with an important subject, and hence it can be of interest to the readership of Plants.
Specifics:
Abstract - A few lines to describe the experimental procedure should be added.
Methods:
𝑅𝑊𝐶 = Apparently, this formula needs editing. See the paper by Fernando (2023, doi: 10.1590/ S1678-3921.pab2023.v58.03360)
See also Ls (equation 3):
Gas – exchange measurements: It is not clear how these measurements were made: for instance, Light conditions, [CO2] and temperature during measurement should be provided.
Fluorescence measurements: It is not clear how these measurements were made. For instance, light saturation pulse to measure Fm (light intensity) and exposure time should be provided.
Page 13: =… shoots were selected, and the gas exchange parameters were determined using a Li-6800 photosynthesizer (Beijing LigaoTai Technology, Beijing, China)=
Li-6800 – Portable Photosynthesis System - The reviewer suggests reviewing this sentence.
= Stomatal opening ( Sow) was measured based on the photographs=
It is not clear how stomatal opening was measured. Further description is needed ( the unit is missing).
Results:
Table 1. Effects …
Growth parameters: The reviewers suggests providing the relative growth rates, instead of just diameter or height increment.
Fig.7 : caption:= Correlation between Sp and photosynthetic parameters. The red lines represent the best-fit linear regressions=
Regression or correlation?
Figures: the quality of figures should be improved.
For instance, Fig. 3H is rather difficult to follow.
Discussion: The discussion´s section needs further improvements.
See for example:
=Stomata control transpiration because they are channels through which plants ex-change gases with the environment; their opening and closing directly determine the photosynthetic rate of plants [30]. Plants can cope with drought-induced damage by maintaining water balance through stomatal transpiration [31].=
The reviewer suggests to move these sentences to the Introduction´s section.
= Therefore, we suggest that exogenous SA improves the drought tolerance of T. ciliata by promoting photosynthesis via two pathways: regulation of photosynthetic parameters and restoration of the activity of the PSII reaction centre =
At the end of this sentence: References should be added.
Interpretation of PCA is missing, see for instance: http://dx.doi.org/10.15517/rbt.v69i2.44489 .
Author Response
Dear Reviewer:
Thank you for your letter and for the reviewers’ comments concerning our manuscript entitled “Salicylic Acid Modulates the Osmotic System and Photosynthesis to Enhance the Drought Tolerance of Toona ciliata" (ID: plants-2750299). Those comments are all valuable and very helpful for revising and improving our paper, as well as the important guiding significance to our researches. We have studied comments carefully and have made correction (highlighted) which we hope meet with approval.
Responds to the reviewer’s comments:
Specifics:
Abstract - A few lines to describe the experimental procedure should be added.
Reply: Thank you for your comment. We have modified the abstract.
Methods:
??? = Apparently, this formula needs editing. See the paper by Fernando (2023, doi: 10.1590/ S1678-3921.pab2023.v58.03360)
Reply: Thanks to your comment, we have revised the formulae in the paper. We apologise for writing leaf water content (LWC) as leaf relative water content (RWC) due to our negligence. Thank you again for your understanding and valuable comments.
See also Ls (equation 3)
Reply: Thanks to your comment, we have revised the formulae in the paper.
Gas–exchange measurements: It is not clear how these measurements were made: for instance, Light conditions, [CO2] and temperature during measurement should be provided.
Reply: Thank you for your comment. We have added detailed parameters in the paper.
Fluorescence measurements: It is not clear how these measurements were made. For instance, light saturation pulse to measure Fm (light intensity) and exposure time should be provided.
Reply: Thank you for your comment. We have added detailed parameters in the paper.
Page 13: =… shoots were selected, and the gas exchange parameters were determined using a Li-6800 photosynthesizer (Beijing LigaoTai Technology, Beijing, China)
Li-6800 – Portable Photosynthesis System - The reviewer suggests reviewing this sentence.
Reply: Thanks for the comment, we have reworked the sentence.
Stomatal opening (Sow) was measured based on the photographs
It is not clear how stomatal opening was measured. Further description is needed (the unit is missing).
Reply: Thank you for your suggestion. We have added the method of measuring the Stomatal opening width (Sow) in our paper.
Results:
Table 1. Effects …
Growth parameters: The reviewers suggests providing the relative growth rates, instead of just diameter or height increment.
Reply: Thanks for your suggestion, we have changed the data in Table 1 to the growth rate of Toona ciliata.
Fig.7: caption: Correlation between Sp and photosynthetic parameters. The red lines represent the best-fit linear regressions
Regression or correlation?
Reply: Thank you for your suggestion, the red line indicates the trend line of the linear correlation model. We apologise for this mistake and have revised the paper.
Figures: the quality of figures should be improved.
For instance, Fig. 3H is rather difficult to follow.
Reply: Thanks for the suggestion, we have reworked the image.
Discussion: The discussion´s section needs further improvements.
See for example:
Stomata control transpiration because they are channels through which plants exchange gases with the environment; their opening and closing directly determine the photosynthetic rate of plants [30]. Plants can cope with drought-induced damage by maintaining water balance through stomatal transpiration [31].
The reviewer suggests to move these sentences to the Introduction´s section.
Reply: Thank you for your suggestion, we have changed the discussion section as you requested.
= Therefore, we suggest that exogenous SA improves the drought tolerance of T. ciliata by promoting photosynthesis via two pathways: regulation of photosynthetic parameters and restoration of the activity of the PSII reaction centre =
At the end of this sentence: References should be added.
Reply: Thank you for your comment. We have revised the introduction and discussion sections and added new references.
Interpretation of PCA is missing, see for instance: http://dx.doi.org/10.15517/rbt.v69i2.44489 .
Reply: Thank you for your comment, we have added some key explanations to the PCA analysis based on the reference you provided.

Reviewer 3 Report
Comments and Suggestions for Authors
1. Figure 1A: How many replicates were used in this experiments? The variance is very high in MD+SA conditions. The experiment needs to be repeated.
2. What does the Sp measurement signify? There is no explanation about it in results.
3. Figure 2F: the MD and MD+SA shows morphological changes but it is still very distinct from CK. To what extent does the SA contributes for rescuing this phenotype? Could there be other molecules that have additive effect? Also, explain some characteristics that scientists are specifically looking for when discussing the morphology.
4. What all the parameters discussed in section 2.3.1 represent? This needs to be explained along with their significance.
5.Figure 3H: Not sure what these micrographs signify. if stomata size is the focus here please find a way to measure that or get more zoomed pictures so that readers can see the difference clearly.
6. The model figure is great but the data supporting the model is not presented in the best way.
Comments on the Quality of English Language
N/A
Author Response
Dear Reviewer:
Thank you for your letter and for the reviewers’ comments concerning our manuscript entitled “Salicylic Acid Modulates the Osmotic System and Photosynthesis to Enhance the Drought Tolerance of Toona ciliata" (ID: plants-2750299). Those comments are all valuable and very helpful for revising and improving our paper, as well as the important guiding significance to our researches. We have studied comments carefully and have made correction (highlighted) which we hope meet with approval.
Responds to the reviewer’s comments:
- Figure 1A: How many replicates were used in this experiments? The variance is very high in MD+SA conditions. The experiment needs to be repeated.
Reply: Thank you for your comments. Fifteen replicates were used in this experiment. We strongly agree with you regarding the variance in the data from the MD+SA conditions you mentioned, however, we apologize that we are unable to supplement the experiment. This experiment was conducted in the summer of 2020, whereas it is now wintertime when an equivalent environment for the experiment is no longer available, and conducting a supplemental experiment at this time would result in test conditions for chlorophyll that are inconsistent with the other treatments. After reviewing the data, we found an anomaly in the MD+SA group, which we have removed. Again, we apologize for our oversight and thank you for your understanding and comments.
- What does the Sp measurement signify? There is no explanation about it in results.
Reply: Thank you for your comments. Sp is the main osmotic regulator in plant cells. After exogenous application of SA, the Sp of Toona ciliata was significantly increased, and the balance of the plant intracellular environment was restored. We explained the significance of Sp in our results and thank you again for your suggestion.
- Figure 2F: the MD and MD+SA shows morphological changes but it is still very distinct from CK. To what extent does the SA contributes for rescuing this phenotype? Could there be other molecules that have additive effect? Also, explain some characteristics that scientists are specifically looking for when discussing the morphology.
Reply: Thank you for your comments. We strongly agree with you that SA can restore mesophyll cell morphology to some extent but it is still very different from CK. We believe that drought may lead to the accumulation of reactive oxygen species in plants. SA does not remove reactive oxygen species in a timely manner, resulting in the failure of mesophyll cells to recover significantly in the short term. We supplement the instructions in the paper according to your requirements and add literature references.
- What all the parameters discussed in section 2.3.1 represent? This needs to be explained along with their significance.
Reply: Thanks for your comment, we have added explanations of all parameters in section 2.3.1.
- Figure 3H: Not sure what these micrographs signify. if stomata size is the focus here please find a way to measure that or get more zoomed pictures so that readers can see the difference clearly.
Reply: Thank you for your comments. The micrographs were intended to visualize the regulatory effect of exogenous SA on the stomata of Toona ciliata under drought stress. Figure 3G is the change in stomatal opening width that we measured. We have followed your comments and partially enlarged Figure 3H so that readers can clearly see the differences. Thanks again for your suggestion.
- The model figure is great but the data supporting the model is not presented in the best way.
Reply: Thank you for your comment. We have revised the model figure.

Round 2
Reviewer 1 Report
Comments and Suggestions for Authors
The paper is OK now, and it can be accepted.
Author Response
Dear Reviewer:
Thank you very much for your time and effort in reviewing my manuscript. We appreciated the valuable comments and suggestions provided, which helped improve the quality of our paper.
Best regards,
Yumin Liu
Reviewer 2 Report
Comments and Suggestions for Authors
I Recommend to revise this sentence
=After 10 days of drought stress, chlorophyll fluorescence parameters were measured using the Li-6800 Portable Photosynthesis System (Beijing LigaoTai Technology, Beijing, China)=
Is the Li-6800 Portable Photosynthesis System the same instrument manufactured by Li-Cor in Nebraska, USA?
If so, I suggest to alter to "using a Portable Photosynthesis System (LI-6800, Li-Cor , Lincoln, NE, USA)" or
"using a Portable Photosynthesis System (LI-6800, Li-Cor , USA, manufactured in China by xxx)"
Author Response
Dear Reviewer:
Thank you very much for your comments, we have revised this sentence to"using a Portable Photosynthesis System (LI-6800, Li-Cor, USA, manufactured in China by LigaoTai Technology)" , and mark it in red in the paper.
Best regards,
Yumin Liu
Reviewer 3 Report
Comments and Suggestions for Authors
N/A
Author Response

(The authors gave the same response as above.)
